# A Comprehensive Grading System for a Magnetic Sentinel Lymph Node Biopsy Procedure in Head and Neck Cancer Patients

**DOI:** 10.3390/cancers14030678

**Published:** 2022-01-28

**Authors:** Eliane R. Nieuwenhuis, Barry Kolenaar, Jurrit J. Hof, Joop van Baarlen, Alexander J. M. van Bemmel, Anke Christenhusz, Tom W. J. Scheenen, Bernard ten Haken, Remco de Bree, Lejla Alic

**Affiliations:** 1Magnetic Detection and Imaging Group, Technical Medical Centre, University of Twente, 7522 NB Enschede, The Netherlands; e.r.nieuwenhuis@utwente.nl (E.R.N.); a.christenhusz@utwente.nl (A.C.); b.tenhaken@utwente.nl (B.t.H.); 2Department of Maxillofacial Surgery—Head and Neck Surgical Oncology, Medisch Spectrum Twente, 7512 KZ Enschede, The Netherlands; b.kolenaar@utwente.nl; 3Department of Radiology, Medisch Spectrum Twente, 7512 KZ Enschede, The Netherlands; J.Hof@mst.nl; 4Laboratorium Pathologie Oost Nederland, 7555 BB Hengelo, The Netherlands; J.vanBaarlen@labpon.nl; 5Department of Otorhinolaryngology—Head and Neck Surgical Oncology, Medisch Spectrum Twente, 7512 KZ Enschede, The Netherlands; A.vanBemmel@mst.nl; 6Department of Surgery, Medisch Spectrum Twente, 7512 KZ Enschede, The Netherlands; 7Department of Medical Imaging, Radboud University Medical Center, 6525 GA Nijmegen, The Netherlands; Tom.Scheenen@radboudumc.nl; 8Department of Head and Neck Surgical Oncology, University Medical Center Utrecht, 3584 CX Utrecht, The Netherlands; R.deBree@umcutrecht.nl

**Keywords:** oral cancer, sentinel lymph node, magnetic tracer, superparamagnetic iron oxide (SPIO), tracer distribution, MRI, lymphography, histopathology, grading system

## Abstract

**Simple Summary:**

With 30% of clinically negative early-stage oral cancer patients harboring occult metastasis, an accurate staging of metastatic lymph nodes (LN) is of utmost importance for treatment planning. A magnetic sentinel lymph node biopsy (SLNB) procedure is offered as an alternative to conventional SLNB in oral oncology, however, a grading system is missing. A proper grading system is preferred to connect the different components of the magnetic SLNB: preoperative imaging, intraoperative detection, and histopathological examination of sentinel lymph nodes (SLNs). This study aims to provide a first grading system based on the distribution of a magnetic tracer, by means of preoperative magnetic resonance imaging (MRI), intraoperative estimation of iron content, and histopathological assessment of resected nodes. Pre- and post-operative MRI and harvested SLNs of eight tongue cancer patients with successful magnetic SLNB procedure were used for analyses.

**Abstract:**

A magnetic sentinel lymph node biopsy ((SLN)B) procedure has recently been shown feasible in oral cancer patients. However, a grading system is absent for proper identification and classification, and thus for clinical reporting. Based on data from eight complete magnetic SLNB procedures, we propose a provisional grading system. This grading system includes: (1) a qualitative five-point grading scale for MRI evaluation to describe iron uptake by LNs; (2) an ex vivo count of resected SLN with a magnetic probe to quantify iron amount; and (3) a qualitative five-point grading scale for histopathologic examination of excised magnetic SLNs. Most SLNs with iron uptake were identified and detected in level II. In this level, most variance in grading was seen for MRI and histopathology; MRI and medullar sinus were especially highly graded, and cortical sinus was mainly low graded. On average 82 ± 58 µg iron accumulated in harvested SLNs, and there were no significant differences in injected tracer dose (22.4 mg or 11.2 mg iron). In conclusion, a first step was taken in defining a comprehensive grading system to gain more insight into the lymphatic draining system during a magnetic SLNB procedure.

## 1. Introduction

In oral cancer, the presence of cervical lymph node (LN) metastases is one of the most important factors for prognosis [1]. Therefore, the detection of metastatic LN is important for treatment planning [2,3]. However, in 30% of early oral cancer (cT1/T2) patients, metastases are not identified during clinical examination or by diagnostic imaging modalities [4,5,6]. To identify these occult metastases, a sentinel lymph node biopsy (SLNB) procedure can be performed [7,8]. The sentinel lymph node (SLN), which is the first draining lymph node, represents the lymphatic status of regional LNs; if the SLN is found to be tumor negative then regional lymph nodes are considered negative too. The conventional SLNB procedure utilizes peritumorally administered radioisotopes for preoperative localization of SLNs by lymphoscintigraphy and for intraoperative detection of SLNs by a gamma probe. After SLNs are harvested, the nodal status is assessed by histopathological examination. The conventional procedure requires strict regulation concerning radioisotope production, transport, and usage [9]. It would therefore be beneficial to have a non-ionizing alternative.

A magnetic SLNB is proposed as an alternative for a radioactive SLNB in, e.g., breast cancer [10,11], prostate cancer [12], and, more recently, for thyroid carcinoma [13] and oral cancer [14,15]. The magnetic SLNB utilizes a magnetic tracer (superparamagnetic iron oxide (SPIO) nanoparticles) and a magnetometer. Similar to the conventional SLNB procedure, a magnetic tracer is peritumorally administered and drains via lymphatic vessels, freely or by macrophages, to the SLNs. In contrast to a radioactive tracer, a magnetic tracer does not have a short half-life time which is advantageous for an SLNB as it provides great freedom in surgical planning. Injected tracer can be used for preoperative SPIO-enhanced magnetic resonance imaging (MRI) to visualize and identify the location of LNs with iron uptake [16,17]. Simultaneously, SPIOs are also used for intraoperative detection of SLNs by a magnetometer [10,11,12,13,14,15]. What should be considered is the difference in SPIO-sensitivity between MRI and the magnetometer [18]. Consequently, the dose of the injected magnetic tracer should lead to a sufficient signal to be detected by a magnetometer without large signal voids on MRI affecting their examination [19].

Since a magnetic SLNB procedure in oral oncology was recently shown to be feasible [14,15], it raises the question of whether it is possible to develop a comprehensive grading system. This grading system could be based on a combination of preoperative MRI, intraoperative LN detection and post-operative histopathology. In the complex anatomy of the head and neck area, a grading system should be of help in discriminating SLNs from higher echelon nodes and determining the effect of signal voids in SPIO-enhanced MRI [15]. Furthermore, determination of the amount of iron within excised magnetic SLNs gives insight into what is still detectable and how much of the injected dose drains to SLN(s). Lastly, the iron content in SLNs, based on counts of the magnetometer, can be related to visual histopathology grading of the iron content.

With the abovementioned components of a magnetic SLNB procedure, we propose a first step towards a comprehensive grading system. Therefore, this study reports the iron distribution to (S)LNs and grading based on preoperative SPIO-enhanced MRI, intraoperative detection by a magnetometer, and histopathology of resected LNs in oral cancer patients who successfully underwent a magnetic SLNB procedure.

## 2. Materials and Methods

Our grading system is based on a selection of patients (*n* = 8/10) with oral squamous cell carcinoma (OSCC), recruited for a feasibility study on a magnetic SLNB procedure (NL6656, Netherlands Trial Register) in the period February 2018–December 2019 at Medisch Spectrum Twente [15]. To assess iron distribution in SLNs, based on SPIO-enhanced MRI, magnetometer counts and histopathological analysis, only patients with a successful magnetic SLNB procedure were selected. All patients were clinically diagnosed with T1-T2N0M0 and scheduled for resection of the primary tumor (border of the tongue) and consecutive elective neck dissection (END) level I-III. Patients were three females and five males, with a mean age of 67 years (range 43–77 years). The SPIO nanoparticle tracer used was Sienna+^®^ [28 mg iron/mL] (Endomagnetics Ltd., Cambridge, UK), which was submucosally injected around the tumor within 24 h prior to surgery. A total volume of 0.4 mL in three patients and 0.8 mL in five patients (corresponding to an iron dose of 11.2 mg and 22.4 mg, respectively) was administered in 4 aliquots. Prior to entering the feasibility study, all patients provided oral and written informed consent. Ethical approval was given by the local medical ethics committee, METC Twente.

### 2.1. Magnetic Resonance Imaging Grading

To identify iron-containing LNs, a preoperative SPIO-enhanced MRI (1.5T, Ingenia, Philips Medical Systems, Best, the Netherlands) was acquired shortly after tracer injection, as described above (range: 22–101 min). Additionally, a postoperative SPIO-enhanced MRI was performed (range 23–49 days post injection) to observe the influence and distribution of SPIOs on MRI during follow-up. The following MRI sequences were acquired in transversal plane of the entire neck using a 16-channel dedicated head and neck coil: T1-weighted (T1w) 3D fast field echo (TR/TE = 25/4.6 ms, flip angle 30°, voxel size 0.75 mm × 0.75 mm × 1.6 mm, FOV = 251–269);T2*-weighted (T2*w) fast field echo (TR/TE = 1700/18.41 ms, flip angle 18°, voxel size 0.62 mm × 0.62 mm × 3 mm, FOV = 247–267).

To evaluate the effect of SPIOs in LNs on SPIO-enhanced MRI, a qualitative five-point grading scale was developed based upon T1w MRI. T2*-weighted MRI was used to confirm the presence of iron scored at T1w sequence. This scale is defined below and illustrated in Figure 1:1: Internal spots of signal void2: Confluence of internal spots of signal void, <25%3: Partial (peripheral) signal void of the node, 25–75%4: Complete signal void of the node, >75%5: Blooming beyond border of the gland

All LNs with iron uptake in level I–III were graded following application of the five-point grading scale by a radiologist experienced in head and neck oncology (J.J.H.).

### 2.2. Amount of Iron in Sentinel Lymph Nodes

Before the SLNB procedure the Sentimag^®^ system was checked for correct operation using a reference sample. SLNs were intraoperatively detected using a Sentimag (Endomagnetics Ltd., Cambridge, UK) magnetometer, within 24 h post injection (range 04 h 06 min–21 h 14 min). One or more magnetically detected SLNs per patient were resected and the amount of iron present in SLNs was determined by ex vivo magnetic readout of, in total, 24 SLNs and the use of a predefined look-up table (LUT). Ex vivo measurements of SLNs were performed, with a handheld probe vertically aligned and the probe tip in an upward position. The Sentimag system was balanced in the air without metal materials in the direct surroundings. When balanced, the SLN was placed at the middle of the probe tip. The following data was recorded per SLN: sensitivity setting of Sentimag, ex vivo Sentimag counts, neck level at which SLN was harvested.

The LUT was generated by evaluating the magnetic readout of ten Sienna+^®^ samples. For each sample, a glass tube with an outer diameter of 8 mm and an inner diameter of 6.5 mm was filled with a Sienna+^®^ dilution up to a volume of 35 µL. Sienna+^®^ was diluted using 0.9% saline and samples contained the following iron doses: 1, 5, 10, 28, 50, 101, 140, 280, 420 and 504 µg. Correct operation of the handheld probe was checked by measuring a reference sample before Sienna+^®^ samples were measured. Prior to each Sienna+^®^ sample measurement, the handheld probe was balanced without metallic materials in the direct surrounding. Measurements were acquired six times per sample, with the sample placed in the middle of and directly next to the probe tip. The handheld probe was held in the same orientation as for the ex vivo SLN measurements. A Styrofoam cubic guaranteed a similar position of each sample to the probe tip. A relation between the amount of iron and corresponding counts was experimentally established and modeled as a first order polynomial by in-house developed software (Matlab environment, R2020a, Mathworks, Inc., Natick, MA, USA), using the six measurements per sample for each sensitivity setting as input. The created LUT is available through 4TU.ResearchData [20]. The first column shows the iron content from 1 µg–500 µg in steps of 0.01 µg. Consecutive columns show corresponding counts for sensitivity settings 1, 2 and 3 of the magnetometer, respectively. For analysis, the iron dose corresponding to magnetic readout per SLN was noted to one decimal place.

### 2.3. Histopathology Grading

SLNs were paraffin embedded in slices of 2 mm, which were histopathologically analyzed using step serial sectioning (five levels with 200 μm interval) and hematoxylin and eosin (H&E) staining. Cytokeratin AE1/3 staining was used to detect metastasis [15]. Iron content in cortical and medullar sinuses was assessed based on H&E stained coupes, using a qualitative five-point grading scale in cortical and medullar sinuses: no iron (grade 0), 1–25% iron (grade 1), 25–50% (grade 2), 50–75% (grade 3), and 75–100% (grade 4) was estimated in the region of interest; an example is given in Figure 2. Each harvested SLN was graded following the five-point grading scale, for one slide, by an experienced pathologist (J.v.B.).

### 2.4. Data Analysis

Grading of SPIO-enhanced MRI and histopathology of LNs was represented as the median (range) for each neck level. The calculated iron content in SLNs based upon LUT was represented as mean ± standard deviation. To compare groups of 22.4 mg and 11.2 mg iron dose injected, a Mann-Whitney U test was performed with confidence interval of 95% (IBM SPSS statistics, version 27). The correlation coefficient (Spearman’s rho SPSS) was determined for iron content and histopathology grading.

## 3. Results

### 3.1. Magnetic Resonance Imaging Grading

For preoperative MRI the total number of identified SPIO-enhanced LNs in level I-III was 71 (90% at ipsilateral site). Eleven of these were identified as most likely SLNs, based on location relative to the tumor and iron susceptibility extent [15]. The highest number of SPIO-enhanced LNs was observed in level II at the ipsilateral side: 54% (38/71), 2(a). These LNs were mainly graded with a higher value compared to LNs in levels I and III, Figure 3b. Level I, II and III were graded (median (range)), respectively: 3.5 (1–5), 4 (1–5) and 3.5 (1–5) at the ipsilateral side and 1 (1–5), 5 (4–5) and 1 (1–1) at the contralateral side. Within preoperative MRI, no significant difference was found at the ipsilateral side, comparing groups with iron doses 11.2 mg and 22.4mg (*p* = 0.24). No LNs with iron uptake were seen at the contralateral side when a dose of 11.2mg iron was administered.

As a result of SLNB and END procedures, most of the identified SPIO-enhanced LNs were removed, however, postoperative SPIO-enhanced MRI still showed areas of susceptibility as a result of iron. These remaining iron deposits were minimally visible on MRI sequences used in daily clinical practice; an example is shown in Figure 4.

When possible, correlation of excised SLNs and preoperative MRI grades of SLNs in most cases showed an MRI-grade 4 or higher. Two of the three identified magnetic SLNs containing metastasis were given an MRI-grade 5 and were assigned as most likely SLNs on MRI. The remaining one was graded 4 or 5, but since three SPIO-enhanced LNs were seen in this level, it was not certain to which the SLN was related.

### 3.2. Amount of Iron in Sentinel Lymph Nodes

The LUT demonstrated, for all three sensitivity settings of Sentimag, a linear relationship between the magnetic readout and the iron content of Sienna+^®^ samples [20]. Of 24 harvested SLNs, 18 were harvested after an injection dose of 22.4 mg iron and six SLNs as a result of an administered dose of 11.2 mg iron. On average, an SLN contained 82 ± 58 µg iron. Based upon iron content, no significant difference was found between the groups with 22.4 mg (81 ± 58 µg) and 11.2 mg (86 ± 64 µg) injected iron dose (*p* = 0.87) (see Figure 5a). On average, SLNs contained 0.36% (22.4 mg) and 0.77% (11.2 mg) iron of the total injected dose. Figure 5b shows, per patient, the number of nodes identified on preoperative SPIO-enhanced MRI, and detected by magnetic detector over time.

In level I, seven (30%) and one (4%) SLN(s) were harvested from the ipsilateral and contralateral sides, respectively. For level II and III, SLNs originated from the ipsilateral side only; these levels contained, respectively, 13 (54%) and three (13%) SLNs. The average iron content of SLNs per level found was: level I: 57 ± 42 µg, level II: 91 ± 61 µg and level III: 114 ± 73 µg. Table A1 illustrates patient characteristics including LN number(s), neck level and iron content.

### 3.3. Histopathology Grading

Of 24 SLNs, three contained metastasis (Appendix A). Regarding iron content grading: level I cortical and medullar sinus were graded as (median) 1.5 (range 0–3) and 3 (0–4), respectively; level II: 2 (1–4) and 4 (2–4); and level III: 4 (2–4) and 4 (3–4). The number of LNs per grade are shown in Figure 6. The three positive SLNs were graded as 2–4, 1–4 and 2–2 for cortical and medullar sinus, respectively. For individual SLNs, the medullar sinus was graded higher in 17 cases, equal in 6 cases and lower in 1 case compared to grading of the cortical sinus (Appendix A). The correlation coefficient of medullar grade and iron content was 0.50 (*p* = 0.013). No significant correlation was found for cortical grade and iron content (0.374, *p* = 0.072).

## 4. Discussion

A comprehensive grading system was proposed in this study, based on iron distribution of (S)LNs in the head and neck area, as a result of magnetic tracer injection for an SLNB procedure. It included analysis of iron-containing LNs on SPIO-enhanced MRI, with SLNs detected by magnetometer and histopathology. Of eight clinically T1-T2N0M0 tongue cancer patients, most draining LNs were located in level II on preoperative MRI and detected by magnetometer. This is in accordance with the preferred draining level of tongue cancers and thus of their potential metastasis [21,22]. Mainly high grades were scored in this level for SPIO-enhanced MRI and medullar sinus on histopathology coupes. A high grade refers to a high iron uptake, which is expected for SLN as the first draining node. 

On average 82 ± 58 µg iron was detected by a handheld magnetic probe in SLNs, regardless of the injected iron dose. This makes the percentage accumulated iron in the SLNs, 0.36% for a 22.4 mg injected iron dose and 0.77% for a dose of 11.2 mg. A study comparing two radioisotopes in oral oncology reported 1.95% and 3.16% as radioactive uptake in SLNs based on lymphoscintigraphy [23]. For breast cancer, tracer drainage to SLN was reported for a magnetic tracer to be 0.3% of the injected iron dose [24], and for radioisotopes this was 0.96% (0.0038–5.14%) of the injected dose [25]. Information on the tracer uptake in SLNs can be used for dose optimization. No significant difference was seen for the different iron doses injected, suggesting that an injected iron dose of 11.2 mg should be enough for the detection of SLNs with potential for less disturbing MRI artefacts [26]. Moreover, no additional clinical benefit is achieved when magnetic tracer deposits are sufficient. It is therefore recommended to not overload SLNs with increased tracer volume [27]. When we take the iron amount of SLNs containing metastasis into account (all found in level II and, in total, only three [15]), the iron amount of these SLNs was, in each case, not the highest among excised SLNs of individual patients. It is known that iron does not locate at sites of tumor cells and fat. In the case of a significant metastasis, this will lead to a reduction in iron capacity or even not show iron at all. However, in the case of clinically node negative, it has been shown that the uptake of SPIO is not influenced by occult metastases [28]. Although the three nodes harboring metastasis did not contain the highest iron content, they met the 10% rule (i.e., at least 10% of the hottest node excised) [7] and belonged to the top three hottest nodes, which is considered clinically sufficient for an SLNB procedure using radioisotopes [29]. One should be aware that this was determined ex vivo, while in vivo magnetic SLNs were searched and might be more difficult to determine if an SLN belongs to the top three hottest nodes. The magnetic readout of the handheld magentic probe is not only determined by the iron content, but by the distance of the source to the probe tip as well. Therefore, preoperative imaging, to indicate the amount of iron, or which LN is most likely to be an SLN, would be beneficial to the surgeon during a magnetic SLNB.

Regarding the SPIO-enhanced preoperative MRI, it was intended to identify LNs with iron uptake. The sequence and timing of MRI and the dose of SPIO are open for optimization, but with available data a grading system was initiated in this study, since a high number of LNs were seen with iron uptake on preoperative MRI [15]. It is expected that not all these LNs showing iron uptake are SLNs, especially since one third of the number was detected by a magnetometer and assigned as SLN. For lymphoscintigraphy, discrimination of SLNs and higher echelons is clear; the number of SLNs is half of the LNs seen with radiotracer uptake [23]. Differentiating SLNs from higher echelon LNs is of utmost importance in a complex anatomy, such as the head and neck [3]. Therefore, a grading system for SPIO-enhanced MRI should evaluate the amount of iron and help in identifying most likely SLNs, so SLNs can be distinguished from other LNs taking up iron. A one-on-one comparison of SPIO-enhanced MRI grading to iron content based on magnetometer readout was not completely possible due to the difference in time and following an END. The influence of time on tracer drainage can be limited by having the SLNB planned directly after SPIO-enhanced MRI. From another perspective, as long as the maximum capacity of iron content in SLN is not reached, the SLN is expected to contain the higher amount of accumulated iron, since the hydrodynamic diameter of SPIO is found to be optimal for an SLNB procedure [30,31]. In cases where a one-on-one comparison of most likely SLNs (including corresponding grade) and SLNs detected by magnetometer (including corresponding iron content) is preferred, it is advised to perform an MRI before and after SLNB (without performing an END). Furthermore, the T2*w sequence can be optimized to support the T1w sequence to identify most likely SLNs. The parameters for T2* were chosen to maximize visualization of the effect of SPIOs. Adjustment of the parameters (including shorter repetition time) may reduce this effect; ideally at T2* only iron uptake in SLNs would be visualized. Postoperative SPIO-enhanced MRI showed iron remainders in LNs resulting from either the time between injection and resection, or from iron remaining at the injection site after tumor resection. Sequences used in daily clinical practice seem to be less influenced by these iron remainders in LNs, due to the use of spin echo sequences which correct field inhomogeneities. Regarding the injected iron dose, a melanoma study showed a significant correlation (r = 0.71, *p* = 0.009) between the injected iron dose and graded MRI artefacts [26]. It also showed that a low tracer dose (0.02 mL Magtrace^®^, 0.6 mg iron) produced intraoperatively detectable SLNs [26].

Histopathology grading of the medullar sinus showed comparable distribution over levels of iron grading when compared to SPIO-enhanced MRI; most LNs in level II were graded high, in contrast to the cortical sinus grading in this same level, which included mainly low grades. This means that more iron accumulated within the medullar sinus, confirming descriptions of iron uptake in SLNs of breast cancer patients and in a porcine model study [27,32]. This suggests that the detectability of SLNs (i.e., iron quantity) is primarily determined by the presence of iron in the medullary sinus. Approaching an LN in vivo or the orientation of an LN during ex vivo measurements on the probe tip should then matter to a lesser degree. This is supportive in detecting SLNs, by determination of the top three LNs with highest iron content. It should be noted, however, that surrounding tissue can influence the magnetic readout, due to its diamagnetism.

The main limitations of this study are the small cohort, the differences in time between procedural steps, and the differences in injected magnetic tracer amount. Consequently, a larger (multi-center) cohort is recommended. This would also enable evaluation of (1) iron dose injected, and (2) time between tracer injection and intraoperative SLN detection on the number of nodes detected and graded. Based on information from this study, an administered iron dose of 11.2 mg found fewer identified (S)LNs per patient and (S)LNs were limited to the ipsilateral side on MRI and for SLNB; however, the iron content and iron grading distribution did not appear different. Furthermore, it was difficult in retrospect to directly link LNs visualized by SPIO-enhanced MRI to harvested and histopathologically examined SLNs. It would therefore be recommended to limit the time between preoperative MRI and an SLNB procedure for research purposes.

## 5. Conclusions

In conclusion, a first step was taken in defining a comprehensive grading system, where data for SPIO-enhanced MRI, ex vivo magnetometer readouts of harvested SLNs, and histopathologic assessment were used to gain more insight into the lymphatic draining system during magnetic SLNB. A direct link between harvested SLNs to preoperative visualized LNs on MRI was difficult to make. SLNs contained, on average, 82 ± 58 µg iron based upon ex vivo magnetometer readouts. No significant differences were found between injection groups (11.2 mg and 22.4 mg iron dose) regarding grading of LN on preoperative SPIO-enhanced MRI, iron content and histopathologic grading.

## Figures and Tables

**Figure 1 cancers-14-00678-f001:**
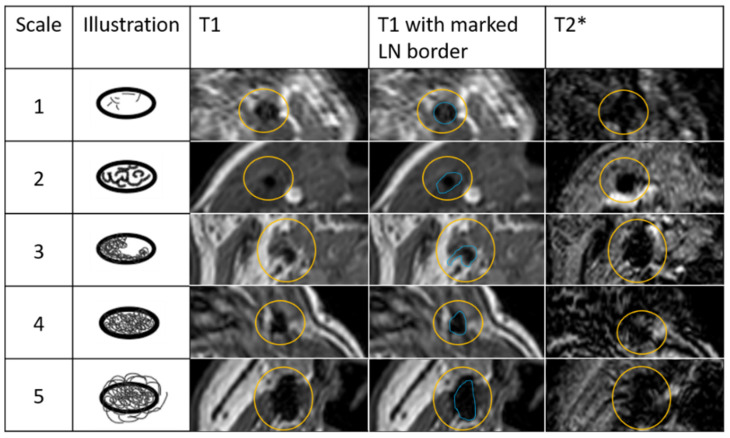
Qualitative five-point grading scale to assess effect of iron uptake in lymph nodes (LNs). Scale: 1—Internal spots of signal void; 2—Confluence of internal spots of signal void, <25%; 3—Partial (peripheral) signal void of the node, 25–75%; 4—Complete signal void of the node >75%; 5—Blooming beyond border of the gland. Column T1 shows examples of LNs with iron uptake. The yellow circle shows corresponding areas in T1 and T2* of the LN with iron uptake. In column T1 with marked LN border, blue line represents LN border.

**Figure 2 cancers-14-00678-f002:**
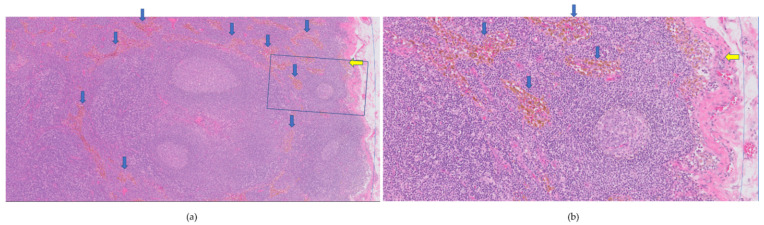
Hematoxylin and eosin-stained coupe of magnetic sentinel lymph node (**a**) 50× and (**b**) 200× magnification, (**b**) is the squared blue box in (**a**). The cortical sinus is shown at the right side (thin blue line) in (**a**,**b**). The iron content was graded as 1 (0–25%), see yellow arrow for focally located iron in cortical sinus. The medullar sinus was scored with grade 4 (75–100%), see blue arrows pointing at iron uptake.

**Figure 3 cancers-14-00678-f003:**
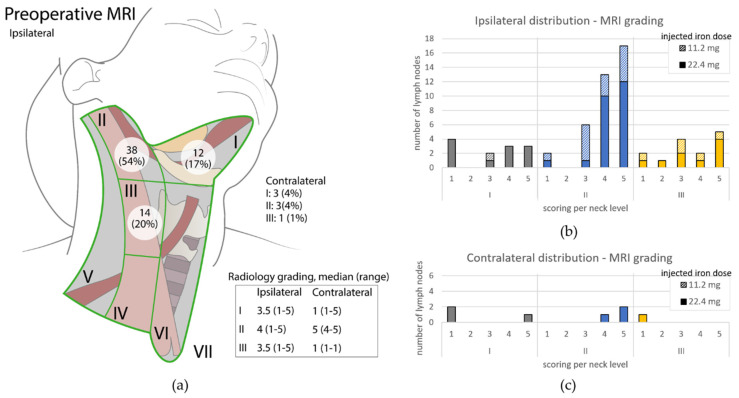
(**a**) Schematic overview of ipsilateral neck. Per neck level the amount of identified SPIO-enhanced LNs and grading on preoperative MRI, median (range) is given. Figure adapted with permission of original authors from [15]; (**b**) Overview of graded LNs at ipsilateral side of SPIO-enhanced preoperative MRI. Per neck level (I = grey, II = blue and III = yellow), the total number of LNs is given based upon their MRI grading score. Fully colored part of the bar represents the identified number of LNs when an iron dose of 22.4 mg was administered; diagonally striped part of the bar includes the LNs graded for patients who received an iron dose of 11.2 mg; (**c**) Overview of graded LNs distributed at contralateral site of SPIO-enhanced preoperative MRI. Description as for (**b**).

**Figure 4 cancers-14-00678-f004:**
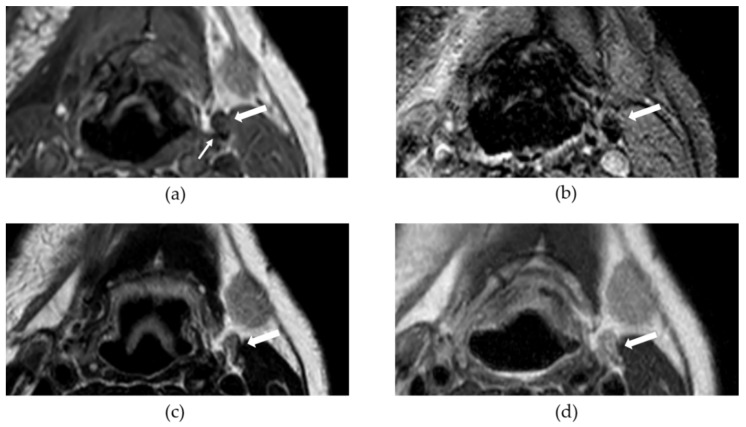
This figure shows an example of an iron containing lymph node (LN) at postoperative MRI for two superparamagnetic iron oxide nanoparticle (SPIO-) enhanced sequences (**a**,**b**) and two sequences for daily clinical practice (**c**) and (**d**). The big white arrow in each subfigure points to a LN (level IIa, left side) with SPIO uptake; the small white arrow in (**a**) points at part of LN with SPIO accumulation; (**a**) T1w 3D FFE (T1-weighted 3-dimensional fast field echo); (**b**) T2*w FFE (T2*-weighted fast field echo); (**c**) T2w SE (T2-weighted spin echo); (**d**) T2w Dixon–in-phase.

**Figure 5 cancers-14-00678-f005:**
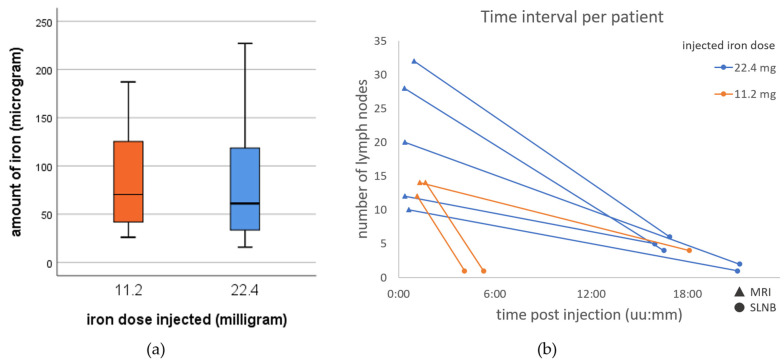
(**a**) Boxplot representing iron content of all detected SLNs for 11.2 mg (*n* = 6) and 22.4 mg (*n* = 18) iron dose injected, no significant difference; (**b**) Per patient the number of lymph nodes showing iron uptake on preoperative magnetic nanoparticle enhanced (triangles) and detected during sentinel lymph node biopsy (SLNB, circles); post magnetic tracer injection is shown in hours:minutes (uu:mm). Orange and blue, respectively, represent patients with 11.2 mg and 22.4 mg iron dose injected.

**Figure 6 cancers-14-00678-f006:**
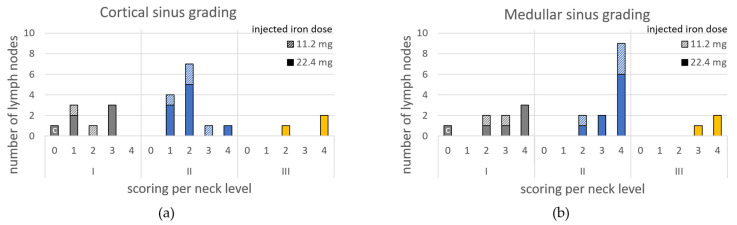
The total number of SLNs per histological iron content grade for each level (I: grey, II: blue and III: yellow), (**a**) cortical sinus and (**b**) medullar sinus. The columns represent the total group, whereas subgroups are identified by diagonally striped part (11.2 mg iron dose injected) and full colored part (22.4 mg iron dose injected).

## Data Availability

The look-up table is available as a dataset via 4TU.ResearchData. Doi: 10.4121/15169674.

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
