# Peer review of "A Comprehensive Grading System for a Magnetic Sentinel Lymph Node Biopsy Procedure in Head and Neck Cancer Patients"

_cancers, 2022, doi:10.3390/cancers14030678_

Round 1
Reviewer 1 Report
This is an interesting work, but some considerations should be made:
- magnetic tracer for SLNB has been used in other pathologies that can be consider closer in its lymphatic drainage to tongue cancer than breast or prostate cancer, for instance in cutaneous melanoma.
- the parameter "blooming" to define the effect of the tracer in nodal imaging is too unspecific, subjective, and in my opinion, difficult to be used. The authors should define in a more strict way the difference between "partly" and "more enhanced".
- the authors should describe the number of excised nodes in the elective neck dissection in each case (perhaps in the table, with the number of isolated SLN).
Author Response
A Comprehensive Grading System for a Magnetic Sentinel Lymph Node Biopsy Procedure in Head
and Neck Cancer Patients
E.R. Nieuwenhuis et al. - Manuscript ID: cancers-1525583
We would like to thank the reviewer for his/her time and the insightful comments. In this
resubmission, we have revised the paper substantially based on the review comments. Below is the
detailed revision statement with point-by-point → response to the reviewer. All remaining minor
issues are corrected in the amended manuscript. Changes were kept by using track-changes as
requested by the editor.
Response to comments by the first reviewer.
Rev. 1
1. Magnetic tracer for SLNB has been used in other pathologies that can be consider closer in its
lymphatic drainage to tongue cancer than breast or prostate cancer, for instance in
cutaneous melanoma
→ To the suggestion of including other pathologies than breast and prostate cancer, we included
a results of a SPIO melanoma study into the discussion regarding use of SPIO dose.
Line 328: “Regarding injected iron dose, a melanoma study showed significant correlation (r
= 0.71, p = 0.009) between the injected iron dose and manually graded MRI artefacts [26].
They also showed that a low tracer dose (0.02 ml Magtrace ®, 0.6 mg iron) produced
intraoperatively detectable SLNs [26].”
2. the parameter "blooming" to define the effect of the tracer in nodal imaging is too
unspecific, subjective, and in my opinion, difficult to be used. The authors should define in a
more strict way the difference between "partly" and "more enhanced".
→ We do agree with the reviewer regarding this matter, therefore we changed this parameter
description, see section 2.1.
Line 124:
• 1: Internal spots of signal void
• 2: Confluence of internal spots of signal void, <25%
• 3: Partial (peripheral) signal void of the node, 25-75%
• 4: Complete signal void of the node, >75%
• 5: Blooming beyond border of the gland
3. The authors should describe the number of excised nodes in the elective neck dissection in
each case (perhaps in the table, with the number of isolated SLN)."
→ Thank you for the suggestion, we have put the number of total resected LNs (including SLNs,
thus result of END and SLNB) into the table of Appendix A – Table A1.

Reviewer 2 Report
This is an interesting and generally well written manuscript, and the results are convincing. For cancer diagnosis and treatment, it is important to create a grading system which will allow proper classification of lymph nodes. Therefore, provide better identification of the sentinel lymph node which represents the regional lymphatic status and allow for early detection of metastasis. This method has advantages over the radioactive tracer, like longer tissue lifetime and no radiation. However as pointed out by the author the time between tracer (SPION) injection, the MRI exam and the sentinel lymph node biopsy is also crucial and needs further investigations to estimate the sentinel lymph node clearance capacity.
Publication of this article will be beneficial for the field. It presents the importance of right approach to achieve the correct diagnosis and treatment of cancer patients.
Specific comments:
- END – abbreviation is missing.
- Can you please provide the micrographs for the histology findings and correlate them with MRI images?
- Appendix A - Table A1 → Interval (hh:mm) → I-MRI2 – what these numbers mean? Shouldn’t it be also hh:mm?
Author Response
A Comprehensive Grading System for a Magnetic Sentinel Lymph Node Biopsy Procedure in Head
and Neck Cancer Patients
E.R. Nieuwenhuis et al. - Manuscript ID: cancers-1525583
We would like to thank the reviewer for his/her time and the insightful comments. In this
resubmission, we have revised the paper substantially based on the review comments. Below is the
detailed revision statement with point-by-point → response to the reviewer. All remaining minor
issues are corrected in the amended manuscript. Changes were kept by using track-changes as
requested by the editor.
Response to comments by the second reviewer.
1. Can you please provide the micrographs for the histology findings and correlate them with
MRI images?
âž” An example of a micrograph of histology finding is added to the method section, figure 2. Since
LNs were not individually scanned in MRI, only in vivo, it was not possible to correlate them with
MRI one to one. However, this would be interesting for further research!
2. Appendix A - Table A1 → Interval (hh:mm) → I-MRI2 – what these numbers mean? Shouldn’t
it be also hh:mm?"
âž” The numbers in Appendix A- Table A1 mean the number of hours. Since, the number of hours for
postoperative MRI is high, the number of minutes became less relevant. In appendix A, this
interval was clarified per column of specific interval now

Round 2
Reviewer 1 Report
I think the manuscript has been sufficiently improved to warrant publication in Cancers.
This manuscript is a resubmission of an earlier submission. The following is a list of the peer review reports and author responses from that submission.
Round 1
Reviewer 1 Report
This is an interesting work, but some considerations should be made:
- magnetic tracer for SLNB has been used in other pathologies that can be consider closer in its lymphatic drainage to tongue cancer than breast or prostate cancer, for instance in cutaneous melanoma.
- the parameter "blooming" to define the effect of the tracer in nodal imaging is too unspecific, subjective, and in my opinion, difficult to be used. The authors should define in a more strict way the difference between "partly" and "more enhanced".
- the authors should describe the number of excised nodes in the elective neck dissection in each case (perhaps in the table, with the number of isolated SLN).